# 30 Years of Improved Survival in Non-Transplant-Eligible Newly Diagnosed Multiple Myeloma

**DOI:** 10.3390/cancers15071929

**Published:** 2023-03-23

**Authors:** Aurelia Chacon, Xavier Leleu, Arthur Bobin

**Affiliations:** 1Hematological Department, University of Poitiers Hospital, 86000 Poitiers, Francexavier.leleu@chu-poitiers.fr (X.L.); 2Service d’Hématologie et Thérapie Cellulaire, PRC, Université de Poitiers, Inserm IC 1402 and U 1313, CHU, 2 Rue de la Milétrie, Cedex, 86021 Poitiers, France

**Keywords:** multiple myeloma, non-transplant-eligible, elderly, immunotherapy, CAR-Ts, bispecific antibodies, monoclonal antibodies

## Abstract

**Simple Summary:**

Multiple myeloma mainly affects the elderly, who are usually not eligible to intensive treatments such as autologous stem cell transplant. Historically, the survival of these patients was short and there were few therapeutic options. Alkylating agents were the main options available until the 2000’s with a relatively low efficacy. However, the last years were marked by the emergence of numerous anti-myeloma drugs, especially with the rise of immunotherapy and targeted treatments. Older patient largely benefited from these advances in the field and therefore their survival has greatly increased. In this review, we summarized the progress made throughout the years in the treatment of older patients affected by multiple myeloma, starting from chemotherapy to innovative immune-based molecules.

**Abstract:**

The treatment of multiple myeloma (MM) has greatly evolved these past few years. Recent advances in therapeutics have largely benefited elderly patients now renamed “non-transplant-eligible” (NTE) patients. Since the 1960s, and for several decades, chemotherapy was the only treatment for MM. Then, the field was marked by the emergence of targeted therapies in the 2000s, such as immunomodulating agents (thalidomide, lenalidomide, and pomalidomide) and proteasome inhibitors (bortezomib, carfilzomib, and ixazomib), which were the first steps towards an increase in survival. Thereafter, the apparition of monoclonal antibodies (mAbs) was considered a milestone in the treatment of MM for both transplant-eligible and NTE patients. Anti-CD38 mAbs can be safely administered to older patients with an impressive efficacy leading to a never-achieved-before survival rate with the triple association of anti-CD38 mAbs, lenalidomide, and dexamethasone. However, progress is still expected with the introduction in the armamentarium for NTE patients of the most recent innovative immunotherapy-based treatments newly introduced in MM, e.g., CAR-T cells and bispecific antibodies. These “improved versions” of immune-based treatments will probably also benefit NTE patients, although further studies will be needed to better understand their role in this population.

## 1. Introduction

Multiple myeloma (MM) is a plasma cell (PC) malignancy characterized by the proliferation of PCs within the bone marrow, usually in association with the production of a monoclonal immunoglobulin (Ig). MM typically occurs among elderly patients. Indeed, the median age at diagnosis is approximately 70 years old, with nearly one-third of patients above 75 years old. The incidence of MM is increasing as the global population is aging. Therefore, more and more patients are categorized as “older” or “non-transplant-eligible” (NTE) when the treatment decision has to be made. However, most of these patients are non-frail and will likely benefit from most of the drug developments made in younger (transplant-eligible) patients.

The elderly MM population is very heterogeneous, with wide differences in age and physical or biological decline features. Older patients are also potentially at higher risk of treatment-related toxicity. Thus, finding the optimal treatment strategy for these patients has long been a challenge. Fortunately, the last 10 years have been marked by tremendous improvements in MM treatment. The emergence of immunotherapy (IT) has profoundly transformed the armamentarium of MM and particularly benefited the elderly because of the highly acceptable safety profile of immune-based treatments.

Naive anti-CD38 monoclonal antibodies (mAbs), the first immunotherapy to demonstrate clinical activity in MM, are now the backbone of treatment for NTE patients upfront. Furthermore, several new immune-based treatments are currently being developed and should be used safely in NTE patients. Among these innovative drugs we can find chimeric antigen receptor T cells (CAR-T cells), bispecific antibodies (BsAbs), and antibody drug conjugates (ADC). Overall, IT, and particularly modern IT, is a shift in the treatment paradigm in MM as it provides effective drugs with enhanced activity and an improved expected safety profile.

Although the future is bright, it is important to remember where we come from in the treatment of NTE patients. Herein, we will describe the progresses and evolution of MM drugs in that setting, which have allowed for improvement in survival rates of older patients affected by MM. It has to be noted that treating such patients could greatly differ across centers, especially before the arrival of anti-CD38 monoclonal antibodies, and notably between Europe and America where, traditionally, oral melphalan was less used. Furthermore, historically, the autologous stem cell transplant (ASCT) criteria can differ greatly from one country to another, and older, fit patients could benefit from ASCT, perhaps by using a reduced conditioning regimen.

## 2. The Beginning: From Alkylating-Based Treatments to Proteasome Inhibitors and IMiDs

The first case of MM was described in 1844, and until the 1960s, the prognosis of patients presenting with MM was abysmal: survival hardly reached a couple of months. In the sixties, alkylating agents were the first treatments to demonstrate anti-tumoral activity in MM, and melphalan thus represented a major breakthrough in the field. At that time, melphalan was the only treatment that could be used, usually given in monotherapy. The efficacy was quite poor, and survival was still very short for MM patients. Corticosteroids were then added to melphalan and rapidly became the gold standard, with a clear improvement in patients’ survival. “Melphalan–prednisone” (MP) served as the basis for many treatment associations for several decades as the associated median overall survival (OS) was 3–4 years. Cyclophosphamide (endoxan) was also rapidly made available and appeared to have significant anti-MM activity either as a metronomic treatment taken on a daily basis or on a weekly basis.

The advent of melphalan was followed by the apparition of immunomodulating agents (IMiDs), including thalidomide, in the 2000s. Thalidomide (T) was previously used in the 1950s against nausea and vomiting during pregnancy, but it was also teratogen, and its use was therefore interrupted in most European countries. It was only years after its anti-angiogenic proprieties were discovered, suggesting a potential effect in MM, that the binding to the cereblon protein was described. This results in the degradation of transcriptional factors (IKZF1, IKZF3) important for MM cells’ survival. Thalidomide was logically investigated in association with MP with hope for an increased efficacy. The IFM 99-06 study [1] compared the combination of MP versus melphalan, prednisone, thalidomide (MPT), and reduced-intensity ASCT for patients between 65 and 75 years old. The MPT association showed the best OS (51.6 months) with no increased toxicities (median OS was 33.2 months for MP, *p* = 0.0006). Consequently, MPT found its way into the first-line treatment of NTE MM as the new standard of care. Of note, patients from the reduced conditioning regimen (100 mg/m^2^) before ASCT had a similar OS to patients from the MP group but with increased toxicities (49% of infection rate vs. 9% for MP), which confirmed that elderly patient should not undergo ASCT. Haematological adverse events (AEs) are more frequent with ASCT. Grade 3 or 4 non-haematological toxic effects were 58% with melphalan/ASCT and 42% with MPT. This study, amongst other, contributed to the definitive interruption of any attempt to propose intensification for elderly MM patients. Since then, the term “NTE patients” came in to replace the term “elderly”.

In a first attempt to demonstrate that frail myeloma patients represented a specific subgroup, the IFM01/01 study [2] evaluated the efficacy of MPT vs. MP in patients >75 years (median age 78.5 with 36% >80). This study also showed the superiority of MPT in terms of survival (median OS 44 months vs. 29.1). Adding the thalidomide to the MP increases peripheral neuropathy (20% vs. 5%) and neutropenia (grade 3/4 23% vs. 9%).

After thalidomide, proteasome inhibitors (PIs), a new class of anti-MM drugs, were successfully introduced to the MM therapeutic landscape. The proteasome plays a major role in protein homeostasis and physiological mechanisms by leading to the degradation of some proteins, including proteins that control apoptosis, cell cycle progression, etc. MM cells have higher levels of proteasome activity; therefore, PIs allow the accumulation of malformed proteins, thus causing cellular apoptosis. Bortezomib (V) was the first-in-class PI to be used in MM. Nevertheless, MP was kept as the backbone for most associations, as in the MPV (melphalan, prednisone, bortezomib) regimen. The phase three VISTA trial [3] studied the association MPV in comparison to MP alone. Bortezomib was administered intravenously. MPV, which led to a median progression-free survival (PFS) of 21 months and a median OS of 56 months (Figure 1), was superior to MP (median OS 43 months, *p* < 0.0001) (Table 1). Interestingly, improvement in OS was seen across all patient subgroups and notably for patients aged ≥75 years. MPV is associated with some important toxicities, even though the sub-cutaneous bortezomib formulation has improved the safety profile, especially peripheral neuropathy because of bortezomib (79% within a median of 1.9 months; in 60% this was resolved within a median of 5.7 months) and the risk of myelodysplastic syndrome because of the prolonged use of melphalan. Thus, MPT and MPV became the new standards of care for years for NTE patients, especially in Europe. However, the main drawback of these two treatments was the limited treatment duration because of the poor safety profile associated with the prolonged drug exposure of both thalidomide and bortezomib in association with MP. This led to the development drugs with a more favorable safety profile in long-term administration in order to avoid early relapse after a fixed-duration first-line.

Second-generation PI carfilzomib (K), with the advantage of less polyneuropathy but increased cardiac risk (cardiac failure, high blood pressure), was also tested for NTE patients [4,5]. This attempt can be considered a failure, as the phase three CLARION trial [6] (KMP vs. VMP) did not demonstrate any superiority in terms of PFS with the use of carfilzomib versus bortezomib (median PFS 22.3 months for KMP vs. 22.1 months with VMP, *p* = 0.906). The utilization of carfilzomib is therefore limited in the NTE population.

In the same period, the IFM group (Institut Francophone du Myélome) and others have studied the role of bendamustine [7] (alkylating agent combined to nucleosidic analogue) with dexamethasone (Bd) and in association with bortezomib and dexamethasone (BVd) for elderly patients. These regimens have never been studied head-to-head with the MP-triplet-based combinations, and although they have been part of the armamentarium in guidelines, they have never been in a position to compete with either MPT or MPV. Currently, bendamustine is not used for NTE newly diagnosed MM (NDMM) and is reserved to patients with few alternative options.

## 3. A Major Step Forward: Lenalidomide-Based Regimens

The apparition of the second generation IMiD, lenalidomide (R) (in 2005), represented a major step forward in the treatment of NTE MM. Lenalidomide is usually better tolerated than thalidomide with decreased risk of neuropathy and can therefore be administered in the long run, upon management of diarrhea and neutropenia. Furthermore, its administration being entirely oral was seen as more convenient than the use of bortezomib, for example.

The phase three FIRST study [8] demonstrated the superiority of the association lenalidomide–dexamethasone (Rd) given until progression over the alkylating agent-based association MPT, which was regarded as one of the gold standards at that time. Indeed, median PFS with continuous Rd was 25.5 months and 21.2 months with MPT (*p* < 0.001) (Table 1; Figure 1). In terms of OS, Rd was also superior, with an improved median OS of 10 months (59.1 months vs. 49.1, *p* = 0.0023). In addition, a lower rate of grade 3 or 4 AEs was observed with Rd (70 vs. 78%), especially fewer neurologic events and secondary cancers.

Although the Rd regimen showed great efficacy and an acceptable safety signature, the triplet associations were already installed in the therapeutic management of MM. Indeed, triplet associations lead to deeper responses, including for high-risk MM patients, and, consequently, to improved survival. The objective was therefore to find the best partner to Rd. The first attempt run by the Italian group GIMEMA (Gruppo Italiano Malattie EMatologiche dell’Adulto) in 2006 with the association of melphalan, prednisone, and lenalidomide (MPR) [9] was unsuccessful given the safety profile issues, mainly cytopenia, that impaired the development of that regimen.

Given that the most efficient drug of the VMP scheme was bortezomib, the next step was therefore to add bortezomib to Rd (VRd) with the hope of deepening responses and increasing survival. The phase three SWOG S0777 trial [10] compared VRd vs. Rd. The experimental arm VRd achieved a median PFS of 43 months versus 30 months in the control group (*p* = 0.0018) (Table 1; Figure 1). The superiority of VRd was also seen in terms of ORR of 82% vs. 72%. However, there were two main limitations in this study: the limited number of patients truly considered as ineligible for transplant as per the worldwide definition (only 43% of patients aged ≥65) and also the rate of grade three AEs, hematological and not hematological, that were more frequent in the VRd arm (82% vs. 75%).

Thus, an adapted VRd version was introduced called “VRd lite” [11] in order to reduce the toxicity associated with this regimen. VRd lite was given over a 35-day cycle (based on the VMP scheme) with oral lenalidomide 15 mg on days 1–21, bortezomib subcutaneous 1.3 mg/m^2^ on days 1, 8, 15, and 22, combined with dexamethasone 20 mg for nine cycles, followed by six cycles of consolidation by VR, similarly to the initial VMP design. Median PFS was 35.1 months (95% CI 30.9; ∞) (Figure 1) and median OS was not reached at a median of follow-up of 30 months. There is also a “VRd lite” version given over a 28-day cycle (as per the Rd scheme) with oral lenalidomide 25 mg on days 1–21, bortezomib subcutaneous 1.3 mg/m^2^ on days 1, 8, and 15, combined with dexamethasone 20 mg for up to six to twelve cycles followed by Rd maintenance treatment.

Ixazomib, a new oral PI, was also added to the bitherapy “Rd” in phase three of the TOURMALINE-MM2 trial [12] (Rd vs. IRd). Ixazomib was given at 4 mg on days 1, 8, and 15 plus Rd at the classical dosage from cycle one to eighteen, then the ixazomib dose was reduced to 3 mg and R to 10 mg until progression or toxicity. Median PFS was 35.3 months with IRd vs. 21.8 months with Rd, and for the high-risk group 23.8 vs. 18 months, respectively. There was no difference between the two groups regarding toxicity; AEs were mostly grade 1 or 2. This entirely oral association can be appealing for NTE patients as it has a limited impact on quality of life with no or few hospitalizations.

## 4. The Rise of Anti-CD38 Monoclonal Antibodies

Immunotherapy is surely one of the most impressive and successful drug developments in the history of MM, first with the discovery of anti-CD38 mAbs after long years of failing attempts to develop a monoclonal antibody for therapeutic use in MM. Abs were developed in the late 1970s after George Kohler and Cesar Milstein invented the hybridoma technology, allowing for the creation of immortal cells producing different antibodies. Thanks to their work, a great range of mAbs were generated. The first mAb to be approved for therapeutic use was the anti-CD20 rituximab in 1997 for the treatment of lymphomas. Logically, rituximab was also investigated, although unsuccessfully, in MM. Many attempts were made to find the right mAb that could allow MM to benefit from the progress of IT, but the search for the optimal target was indeed a long journey.

The first mAb to find its way into MM treatment was the anti-signaling lymphocytic activation molecule 7 (SLAMF7) [13], renamed CS1, elotuzumab. SLAMF7 can be found on different cells whose PCs are natural killer (NK) cells, which are very important for the mechanism of action of elotuzumab. The dual effect of elotuzumab is principally due to the antibody-dependent cellular cytotoxicity (ADCC) on tumor cells, while in parallel activating NK cells that also express CS1 on their cell surface. Lenalidomide was found to exhibit a synergistic activity with elotuzumab, which led to the first approval of a mAb in MM with the association elotuzumab, lenalidomide, and dexamethasone in 2015 for patients with relapse or refractory MM (RRMM), followed by a second approval with the third-generation IMID pomalidomide. Nevertheless, elotuzumab has no clinical activity when used alone, and failed when being developed upfront, particularly in NTE MM.

CD38, first described in 1980, is a glycoprotein that serves as a receptor and an adhesion molecule and also has enzymatic functions. CD38 has a major role for survival of PCs. Daratumumab is the first fully humanized mAb targeting CD38, and it was first approved for RRMM combined with either bortezomib or lenalidomide or in monotherapy. Daratumumab adds an anti-enzymatic activity to the classical mAb mechanisms of action, ADCC, CDC (complement-dependent cytotoxicity), ADPC (antibody-dependent cellular phagocytosis), and apoptosis.

After the promising results of studies for RRMM, daratumumab-based trials were specifically designed for NTE NDMM patients. The phase three ALCYONE trial [14] studied the association dara-MPV vs. MPV; median PFS was higher for patients in the daratumumab group (36.4 vs. 19.3 months, *p* < 0.0001) in the primary analysis (median follow-up of 16.5 months) (Table 1; Figure 1). Moreover, in the last update of the results, with a median follow-up of 74 months, median OS was significantly more important with daratumumab (82.7 vs. 53.6 months, *p* < 0.0001) [15]. In the daratumumab group, 28.3% of the patients achieved minimal residual disease (MRD) negativity (10^−5^ sensitivity threshold) as compared with 7% of the control group (*p* < 0.001). Severe AEs were more frequent in the daratumumab group (82.9 vs. 77.4%), but the rate of discontinuation was lower with dara-VMP (4.9 vs. 9%).

Interestingly, daratumumab was also investigated in parallel with lenalidomide and dexamethasone upfront for NTE patients in the phase three MAIA trial [16,17], Dara-Rd vs. Rd until progression. In total, 368 patients were treated with Dara-Rd, which led to a median PFS never reached before for NTE patients of 61.9 months versus 34.4 months (29.6–39.2 hazard ratio (HR) 0.53 [95% CI 0.43–0.66]; *p* < 0.0001) with Rd (Figure 1) VR. Concerning results of the analysis in subgroups, the experimental group had a better ORR across frailty patients (92.9% vs. 81.6%; *p* = 0.0265) [18] (Table 1). Daratumumab reduced the risk of disease progression or death by 34%, although the median PFS of the control Rd arm was the best ever observed in a registration study with Rd regimen. Furthermore, MRD negativity (10^−5^) in patients <75 years was 36.1% vs. 12% without dara and sustained-MRD negativity at 6 or 12 was higher for DRd, 14.9% vs. 4.3% and 10.9% vs. 2.4%, respectively, which is associated with longer survival. The most common AEs were cytopenia, especially neutropenia (54% rate with Dara-Rd). In the experimental group, a higher rate of infection was demonstrated (pneumonia in 19% of patients with dara vs. 11% without dara).

## 5. Quadruplet-Based Regimens, Utopia, or Future for Non-Frail NTE NDMM?

With the success of daratumumab came the emergence of novel anti-CD38 mAbs. Isatuximab is the second anti-CD38 chimeric mAb for therapeutic use that has been developed. The mechanism of action of isatuximab is slightly different from daratumumab, although experts in the field consider that the two drugs show quite similar clinical activity. It was then tested upfront specifically for NTE patients in the phase three IMROZ study [19] (NCT03319667) Isatuximab-VRd vs. VRd (primary end point median PFS), and in the IFM 2020-05 BENEFIT trial (NCT04751877, Isatuximab-VRd vs. Isatuximab-Rd, (65–79-year-old NDMM NTE patients)), primary end point 10^−5^ MRD negative rate at 18 months, surrogate to sustained MRD (12 and 18 months). The results of the two studies are awaited but read out is expected in 2023. Daratumumab is also being evaluated as part of a quadruplet regimen, as in the phase three CEPHEUS trial [20], among other studies, dara-VRd vs. VRd (NCT03652064).

In the GMMG-CONCEPT [21] phase II trial, they studied the benefit of quadruple therapy (Isa-KRd) in elderly patients with high-risk MM. After a 24-month follow-up, the overall response rate (ORR) is 83%. In terms of toxicity, there are no more AEs.

Physicians are investigating quadruplets for this subset of patient with the objective of achieving deeper responses and longer remissions, and maybe allowing some patient to be treated with only one line of treatment throughout their disease course. The concept of giving four drugs to older patients was only permitted due to the noticeably good safety profile of immune-based treatments. However, compliance to treatment remains a key issue, along with the continuous administration of these drugs. It is expected that some patients might not be fit for such regimens, particularly the frail and intermediate frail NDMM NTE patients. Tailoring treatments to each patient is still a major challenge with the elderly.

## 6. Modern Immunotherapies: Suitable for NTE Patients?

Progress in the field of molecular biology and immunology has allowed us to imagine structural modification and change in specificity of T cells without being limited by the major complex of histocompatibility recognition. CARs are genetically engineered receptors expressed by T cells (or any cells), thus conferring them specific properties, e.g., anti-tumor activity. CAR-Ts were first developed in the early 1990s by Zelig Esshar in Israel but were not clinically active. The first clinical application was made possible in the early 2010s following the work of Carl June in the United States. In 2012, a seven-year-old patient suffering from acute lymphoid leukemia successfully received the first anti-CD19 CAR-T cell [22]. Following this achievement, the CAR technology was logically studied in most hematological malignancies.

In MM, most CAR-Ts are second-generation, i.e., with a co-stimulation molecule (4-1bb or CD28). Two CAR-Ts are advanced in their development, idecabtagene vicleucel (ide-cel) and ciltacabtagene autoleucel (cilta-cel), both targeting B cell maturation antigen (BCMA) expressed by MM cells. BCMA has a role in PC and MM cells’ proliferation and survival. CAR-T cells are now reserved for highly advanced MM, but they should soon benefit patients in early lines of therapy. Ide-cel was evaluated in the phase two KarMMA study [23] (Table 2). In this study, 128 patients with RRMM (median of prior lines = 6) were enrolled, including 45 patients aged 65 years old or more; median age was 61 years (range 43–78 years). Patients were divided into several cohorts with an escalating dose of CAR-Ts infused. The ORR across all cohorts was 73%, with 33% of patients achieving a complete response (CR), and median PFS was 8.8 months IC 95%, 5.6–11.6. Common side effects were mostly neutropenia (91%) and cytokine release syndrome (CRS), more frequently grade 1/2 (84%); 18% of patients had neurological AEs, such as immune cell-associated neurologic syndrome (ICANS). Cilta-cel was studied in the phase two CARTITUDE-1 study [24] where 97 RRMM patients with a median age of 61 years were included (Table 2) (median of prior lines = 6). Of note, the oldest patient was 78 years old. Cilta-cel led to an impressive ORR of 97%, including 82.5% of stringent CR. Furthermore, the rate of MRD 10^−5^ negativity was 92%. The 27-month PFS was 54.9% [25]. Most important non-hematological AEs were CRS in 84% of patients and were grade 3/4 in 4%. Regarding ICANS (16%), 2% of patients had grade ¾ AEs.

CAR-T cells are usually well tolerated or at least their toxicity is now considered predictable and manageable. Indeed, the main risk with the use of CAR-T cells is the CRS. The neurological toxicity associated with CAR-T cells, either CRS/ICANS (usually an early event), or later parkinsonian-like syndrome, is quite rare in the context of anti-BCMA CAR-T cells used in MM but has to be closely monitored. However, long-term complications, such as cytopenia, hypogammaglobinemia, and infections, can also be observed and should be acknowledged during a patient’s follow-up. Given this relatively low toxicity, it is considered possible to administer CAR-T cells to the elderly without a specific limit of age, although this requires careful patient selection, and data focusing on NTE patients are sparse. This is why studies specifically designed for patients who will not have an ASCT are planned, such as CARTITUDE-5 (VRd followed by cilta-cel versus VRd followed by Rd, NCT04923893). Of note, other CAR-T trials will focus on newly diagnosed multiple myeloma and will help define their use in early therapeutic strategies, such as KARMMA-4 (phase one, ide-cel for high risk MM, NCT04196491) and CARTITUDE-6 (dara-VRd followed by cilta-cel or dara-VRd followed by ASCT, NCT05257083).

Amongst the innovative anti-MM therapeutics, BsAbs are gaining interest. BsAbs have two specific antigen binding sites, one to target T cells via CD3 and another to target the antigen on the tumor cell, therefore allowing the proximity of the T cell to the targeted cell for initiating its cytotoxicity. As for CAR-T cells, many BsAbs are anti-BCMA Abs, although new targets are being explored. Unlike CAR-Ts, they are available “off the shelf” and mostly administered subcutaneously, although they usually require weekly administration. Teclistamab, one of the most advanced BsAbs in the field, was studied with the MajesTEC-1 phase one/two study [26]. In total, 159 patients aged between 33 and 84 years were enrolled in the study (Table 2). They had previously received a median of five lines of treatment. First, data with teclistamab are promising because the ORR was 65% and the median PFS was 11.3 months in MajesTEC-1. The most common AEs were CRS (67%), mostly grade 1/2 and neutropenia (grade 3/4 45%). However, a few patients experienced ICANS, with 2.5% fully resolving. The ongoing MAJESTEC-7 (NCT05552222) study will give additional information for patients ineligible for ASCT as it will evaluate the association of teclistamab, daratumumab, and lenalidomide (Tec-DR) versus DRd.

Elranatamab is another anti-BCMA BsAb that led to an ORR of 70% for 58 RRMM patients in the MAGNETISMM-1 study [27] (Table 2). Patients were included in the study without any limit of age; thus, the oldest patient that received the drug was 84 years old. Furthermore, the toxicity profile of BsAbs is nearly similar to the one observed with CAR-T cells; although the severity of CRS is usually higher with CAR-Ts, neurotoxicity is quite rare for both, but the rate of infections of all types (bacterial, viral) is increased with BsAbs compared to CAR-Ts. More BsAbs are emerging with different targets, such as GPRC5D (talquetamab) and FcRH5 (cevostamab), amongst others.

Currently, these novel immune-based treatments are not being used in daily practice for NDMM and especially transplant ineligible NDMM patients. However, unlike ASCT, theoretically they seem suitable for the latter as their safety profile seems manageable for most patients. Careful screening of patients will be required to better determine who can receive this type of treatment without experiencing severe treatment-related toxicities. However, progress is being made in AE management as different strategies are being evaluated, such as the early use of tocilizumab or novel CAR designs. Interestingly, these treatments may also allow us to observe an immunological effect that will probably contribute to achieving MRD negativity and sustained responses, consequently allowing for prolonged survival.

## 7. What about Frail Patients?

In the future, with the constant evolution of the therapeutic landscape and the emergence of highly effective drugs, MM physicians will probably have to remind themselves that the choice of treatment must be guided by features other than expected drugs’ efficacy, particularly for NTE patients e.g., the patient’s characteristics, polypharmacy, expected quality of life, and adherence to treatment. Indeed, this very heterogeneous population usually presents with pre-existing comorbidities, and treatments should ideally be tailored to each patient. Most NTE patients will surely be eligible for innovative treatments, likely after a careful evaluation and with close follow-up, but the frailest ones will not benefit from such therapeutics.

The first issue is to identify those frail patients. Currently, different tools are at the clinician’s disposal, notably several geriatric assessment scores that can be very useful for that purpose, such as the IMWG [28] score and the R-MCI [29]. These scores allow patients to be categorized into three groups: fit, intermediate fit, and frail. Working with an onco-geriatrician is also greatly encouraged in some cases. Only a few studies have focused on patients with either severe comorbidities or who are very elderly, i.e., frail patients. However, they represent a large proportion of patients suffering from MM. For instance, in the phase III MAIA trial, 44% of patients were aged 75 or older. For this subgroup, the objectives of the therapeutics have to be re-defined, especially by considering the patient’s quality of life. The ultimate goal for the frail population would certainly be to be treated with a unique line of treatment, ideally with a fixed duration, with a favorable safety profile allowing a preserved quality of life but without suboptimal efficacy. Therefore, different solutions can be considered. First, fixed-duration treatments seem appealing for these patients in order to limit the risk of cumulative toxicities. As for now, most of the current schema for NTE patients are given in the long term. However, this paradigm could evolve in the years to come. Second, oral treatments appear necessary when considering frail patients so that they will not have to come to the hospital so often. In that context, new-generation PI ixazomib represents a suitable drug for the elderly as it can be used in all-oral combinations. Third, the interruption of dexamethasone responsible for various side effects (high blood pressure, diabetes, etc.) could also improve adherence to treatments for the elderly and limit toxicities. In this way, Larocca et al. [30] evaluated a treatment with either lenalidomide–dexamethasone (Rd) continuously or Rd followed by R maintenance with the discontinuation of dexamethasone. The event free survival (EFS) was increased in the Rd-R group (10.4 vs. 6.9 months) and globally lower rates of AEs were observed. The idea of interrupting dexamethasone for frail MM patients is spreading as it is evaluated in other clinical trials. Indeed, the IFM 2017-03 will analyze the association of daratumumab and lenalidomide (DR) alone versus Rd (NCT03993912). Fourth, limiting AEs is a major issue for frail patients, who have an increased susceptibility to treatment-related toxicity because it is associated with early deaths. Nevertheless, if the reduction of AEs comes with a diminished version of drugs’ association, it is highly possible that the efficacy of such treatments will also be reduced, i.e., undertreating patients. Hypothetically, a specific full-dose line of treatment aimed at frail patients should emerge. Overall, for this subgroup, the challenge remains to find the best treatment with minimal toxicity in order not to jeopardize the clinical benefits; in other words, seeking the right balance between efficacy and toxicity. More specific clinical trials are expected for this population of patients.

## 8. Conclusions

The treatment options for NTE patients have greatly evolved over the past 30 years. Overall, we moved from melphalan-based induction regimens to bortezomib and lenalidomide-based associations and now to anti-CD38 based treatments. For a long time, specific drug developments were missing for elderly patients, but technological improvements in drugs have made them more suitable for these patients, which has led to great successes. In parallel, the growing interest in NTE patients has helped to better understand their specificities and to adapt the objectives of the treatment strategy. Still, some questions are left unanswered, such as the duration of the treatment, which is still a debate among MM physicians. Continuous treatment is currently the standard of care, but the new approved treatments allow deep remissions (with high rates of MRD negativity) similar to those achieved for younger patients, which usually translate into prolonged survival and could therefore question the possibility of developing fixed-duration schemes. Armed immunotherapies, such as CAR-Ts and BsAbs, are making a sensational entrance into MM treatment options and seem fitting for NTE patients. However, their position in the therapeutic strategy of NTE MM is yet to be defined. Should they be used early in the disease course where the immune system is not yet exhausted and has not been harassed by multiple lines of treatment, or later. when few options remain? Future studies will help clarify these points and will undeniably help to better improve the way older patients are treated.

## Figures and Tables

**Figure 1 cancers-15-01929-f001:**
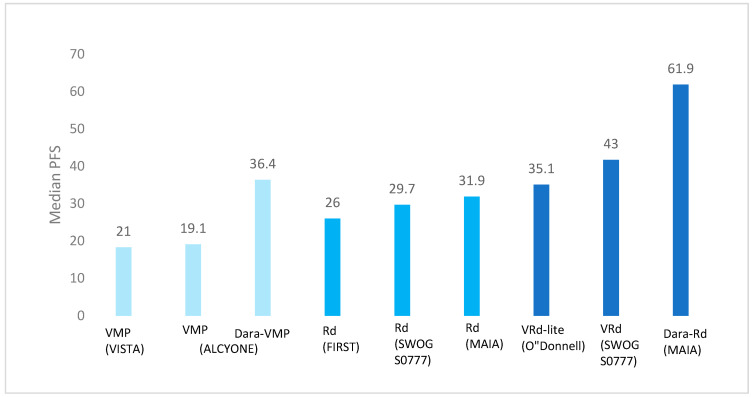
Overview of median PFS in NTE NDMM phase 3 clinical trials.

**Table 1 cancers-15-01929-t001:** Overview of treatment in NTE NDMM.

	FIRST Rd	VISTA VMP	SWOG S0777 VRd	VRd Lite	ALCYONE Dara-VMP	MAIA Dara-Rd
Number of patients enrolled	1623	682	525	53	706	737
Number of patients ≥75 years old	567	216	-	4	211	321
Response rate, n (%):						
▪ ORR	81	71	81.5	86	90.9	92.9
▪ ≥CR	22	30	15.7	32	46	43.6
▪ ≥VGPR	26	-	27.8	66	73	30.8
Median PFS, months	26	21	43	35.1	36.4	61.9
OS, months	59.1	56	75	-	82.7	73.9
MRD negativity rate, n (%)	-	-	-	-	28.3	28.8
Adverse events (%):						
▪ Haematological gr 3-4	30	40	82	14	40	57
▪ Infection gr 3-4	32	-	-	-	30	41.7
▪ Neurological toxicity	-	79	33	<1	-	-

**Table 2 cancers-15-01929-t002:** Data from pivotal CAR-T cells and bispecific antibodies clinical trials.

	CAR-T Cells		Bispecific Abs	
	KarMMa 1 (Ide-cel) n = 128	CARTITUDE-1 (Cilta-cel) n = 97	MagnetisMM-1 (Elranatanab) n = 58	MajesTEC-1 (Teclistamab) n = 159
Age, years (range)	33–78	43–78	42–80	33–84
Median number of prior lines	6	6	6	5
ORR, %	73	97	70	63
Median PFS, months	8.8	54.9	NR	11,3
CRS, %	84	83	87.3	67
CRS grade ¾, %	5	4	0	0.6
ICANS, %	18	16	20	2.5
ICANS grade ¾, %	3	2	0	0

## Data Availability

Data sharing is not applicable to this article.

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
