# Peer review of "30 Years of Improved Survival in Non-Transplant-Eligible Newly Diagnosed Multiple Myeloma"

_cancers, 2023, doi:10.3390/cancers15071929_

Round 1

Reviewer 1 Report

In this paper authors reported the “30 years of improved survival in non-transplant eligible newly diagnosed multiple myeloma”.

 The idea to review the treatment story of multiple myeloma is appropriate based on the consideration that the overall survival of multiple myeloma patients has improved over the years due to the introduction of proteasome inhibitors, immunomodulatory drugs and anti-CD38 monoclonal antibodies.

 Pag. 3. Please add the study TOURMALINE-MM2 concerning the use of ixazomib

 I suggest to give more information concerning the adverse events during the different treatments and the patient’s quality of life.

 In Table 1. “overview of treatment in NTE NDMM” please add also a column with the adverse events

 The English and the general presentation are on the average so the paper is acceptable and justified this work

Author Response

Thank you for your analysis about this paper. 
We hope that the changes will suit you. 

Best regards. 

Reviewer 2 Report

The manuscript is a good review of the field of non-transplant eligible newly diagnosed multiple myeloma patients. The review covers a long period of time, starting with conventional chemotherapy and ending with possible future approaches including T-cell redirection therapy.

However, some corrections need to be made. Moreover, a few areas should be explored to further strengthen the manuscript.

1)    The bibliographic references are not complete and the correspondences are not inserted in the text so they are difficult to follow.

2)    Page 3:  “MPV is associated to some important toxicities especially peripheral neurophaty …”. It would be useful to mentione as the use of less intensive VMP regimens and development of the sub-cutaneous bortezomib formulation have improved the safety profile without sacrificing efficacy.

3)    Page 4: about ALCYONE trial, data on MRD can be reported specifying method and sensitivity, data on OS (update ASH 2022) could be added.

4)    Page 5: about MAIA trial, the sentence “Besides, sustained–MRD negativity at 6 or 12 months increased PFS……” is difficult to understand and needs rephrasing. Moreover updated data on sustained-MRD at 12 and 18 months (ASH 2022) could be added. 

5)    Page 7 (lines 257-271): The number of prior lines of therapy reported in the KarMMA and CARTITUDE trials could be added to contextualize efficacy data. Data on CARTITUDE (PFS) could be update (Martin JCO 2022)

6)    Page 7 (lines 269-270): the authors write that in the CARTITUDE trial “the oldest patient was 68 years old”. This seems not correct as the range was 43-78 (reported by Madduri ASH 2020), while 56-68 is the IQR.  

7)    Page 7: the authors write as “CAR-T cells are usually well tolerated or at least their toxicity is now considered predictable and manageable”. It would be nice to explain that side effects associated with CAR-T cells include both short-term toxicities (as CRS and neurotoxicity) and long-term complications as cytopenia, hypogammaglobinemia and infections. Moreover it would be useful to be underline that data on efficacy and safety in the older population are sparse and a careful selection is required. Main active and planned trials in newly diagnosed MM should be mentioned with a short explanation (eg KarMMA4, CARTITUDE4,  CARTITUDE 2)

8)    Page 7-8: authors write as “the toxicity profile of BsAbs is similar to one observed with CART cells…”. It would be nice if authors could comment better on some difference between CART and BsAb, as difference in the severity of CRS and incidence and severity of neurotoxicity, as well difference in the modality of administration. Main active and planned trials with BsAb in newly diagnosed MM should be also mentioned with a short explanation (eg Majestc-7)

9)    Authors could better underline as CART and BsAb are not in the clinical practice in newly diagnosed patients but they are being studied or will be explored in already planned clinical trials. Even though toxicity related to these novel strategies may be a limitation especially for elderly patients, several approaches are also being developed to improve the safety, (eg incorporation of a suicide gene safety system for CAR-T therapy, early use of tocilizumab..).

10) Table 2: data need to be checked again and some correction are needed. About CARTITUDE-1 1) the age years range reported by MAddure ASH 2020 is  43-78 (56-68 is reported as the IQR), 2) the median number of prior lines is 6 not 3 (Bedeja Lancet 2021) 3) PFS can be reported as % at 27 months  (55%) (Martin JCO 2022), 4) the reported % of 21% is not correct for ICANS (16%) as 21% is related to all neurotoxicities (it should be made explicit); About KarMMa 1) the ORR is 73% (Munshi NEJM 2021); For all trials 1) adding the % of gr 3-4 CRS and ICANS could be useful

Author Response

(The authors gave the same response as above.)

Reviewer 3 Report

  Chacon et al write a well thought out historical summary of the evolution of myeloma treatment for patients traditionally considered transplant ineligible. The authors do a nice job of laying out the major trials that have changed the field in this regard and also present potential new future options including antibody drug conjugates, bispecifics and cell therapies. I very much enjoyed reading about the potential of these new therapies in elderly patients and I agree with the authors in this regard.

Despite this, I believe the authors describe mostly the treatment of patients in France or perhaps in Europe. This should probably be noted somewhere in the paper. For example, in the USA transplant is standard of care for elderly fit patients with these patients being transplanted with dose reduced melphalan.  See one recent example of a publication in about transplant in the elderly in a US center:

https://pubmed.ncbi.nlm.nih.gov/34626863/

Additionally, VMP is not a regimen that has traditionally been used in the USA due to its poor efficacy, in favor of VRd lite for frail elderly patients and now even VRd lite with Dara. Perhaps the author could add somewhere in the manuscript that there are regional differences in the treatment of elderly patients with myeloma.

Finally the authors describe how the use of frontline carfilzomib in the elderly is limited due to the CLARION study not meeting its primary endpoint. The current study GMMG-CONCEPT (NCT0310484) (https://www.nature.com/articles/s41375-021-01431-x) for high-risk patients is accruing elderly patients as well with initial very encouraging results in terms of response underscoring the potential of KRd+cd38 antibody regimens of achieving deep responses including MRD negativity. Thus, it may need to be revised that for fit elderly patients (and especially those with high-risk myeloma) carfilzomib may (or does) have a role.

Author Response

Thank you for your analysis about this paper. 
We hope that the changes will suit you. 

Response to Reviewer 3 Comments

Point 1: Despite this, I believe the authors describe mostly the treatment of patients in France or perhaps in Europe. This should probably be noted somewhere in the paper. For example, in the USA transplant is standard of care for elderly fit patients with these patients being transplanted with dose reduced melphalan.  See one recent example of a publication in about transplant in the elderly in a US center:

https://pubmed.ncbi.nlm.nih.gov/34626863/

Response 1 : Indeed, there are great differences from a country to another. As French clinicians our practice is obviously influenced by European standards but we tried to provide an objective work. Yet, we decided to acknowledged, as you recommended, theses differences in the way of treating elderly patients by modifying the introduction as you will see below.

Line 53-60 : “Although the future is bright, it is important to remember where we come from in the treatment of NTE patients. Herein, we will describe the progresses and evolution of MM drugs in that setting, which allowed to improve the survival of older patients affected by MM. It has to be noted that treating such patient could greatly differ across centers, especially before the arrival of anti-CD38 monoclonal antibodies, and notably between Europe and America where traditionally oral melphalan was less used. Also, historically the ASCT criteria can be really different from a country to another and older fit patients can benefit from it, maybe by using reduced conditioning regimen.”

Point 2: Additionally, VMP is not a regimen that has traditionally been used in the USA due to its poor efficacy, in favor of VRd lite for frail elderly patients and now even VRd lite with Dara. Perhaps the author could add somewhere in the manuscript that there are regional differences in the treatment of elderly patients with myeloma.

Response 2 : We hope that with the changes made in the introduction and the addition of the sentence below you will find the manuscript more balanced between American/World and European practice.

Line 117-118 : “Thus, MPT and MPV became the new standards of care for years for NTE patients, especially in Europe”

Point 3: Finally the authors describe how the use of frontline carfilzomib in the elderly is limited due to the CLARION study not meeting its primary endpoint. The current study GMMG-CONCEPT (NCT0310484) (https://www.nature.com/articles/s41375-021-01431-x) for high-risk patients is accruing elderly patients as well with initial very encouraging results in terms of response underscoring the potential of KRd+cd38 antibody regimens of achieving deep responses including MRD negativity. Thus, it may need to be revised that for fit elderly patients (and especially those with high-risk myeloma) carfilzomib may (or does) have a role.

Response 3: We have added a few data about this study.

Line 129-131 : “In the GMMG-CONCEPT[1], phase II trial, they studied the benefit of quadruple therapy (Isa-KRd) in elderly patients with high-risk MM. After a 24-months follow-up, the ORR is 83%. In terms of toxicity, there is no more AEs.” 

Best regards.

Round 2

Reviewer 2 Report

Some corrections still need to be made.

1)    The bibliographic references are not complete (journal, year, pages etc as requested by the journal)

2)    Please double check the paragraphs added in the review, maybe there are some errors (eg pg 9 line 383 : “there type”)

3)    Page 2 line 66: the acronym ASCT is used for the first time so it needs to be explained (and remove the explanation at line 92)

4)    Page 3 line 98: the line is not readable

5)    Page 3 line 124-125: explain that the data in brackets refer to the VISTA study with bortezomin iv

6)    Page 4 line 189: (95% CI 30.9-….) , a data appears to be missing

7)    Page 6 line 263: please data on MRD best (and not only sustained) can also be reported specifying

8)    Page 3 lines 138-143: it would be more correct to move the comment about the GMMG-CONCEPT trial to the paragraph 4 (antiCD38) or 5 (quadruplet..)

9)    Table 1: specify what the AEs numbers refer to (% ?)

10) The acronym CAR-T is explained twice: pg 2 line 49 and pg 8 line 297

11) Page 8 line 299: “ CAR-T’ s ” maybe it's a typing error (for CAR-T cells or CAR-Ts  ?)

12) The words “plasma cell”, “multiple myeloma” and “autologous stem cell transplant” are sometimes referred to as an acronyms (PC, MM and ASCT) other in full, please check and do just the same

13) Page 8 lines 324/325:  the acronym s CRS and ICANS are used for the first time so they need to be explained (and remove the explanation at lines 334-335/336)

14) Page 9 line 352: the acronym BsAbs has already been explained previously, please remove the explanation

15) Page 9 line 370: the acronym RRMM is used for the first time so it needs to be explained

16) The acronym AEs is used many times but it is never explained

Author Response

Here are the latest changes.

Thanks you for your feedback.
